# Bacillus Calmette-Guérin (BCG) therapy lowers the incidence of Alzheimer's disease in bladder cancer patients

Ofer N. Gofrit[1]*, Benjamin Y. Klein[2], Irun R. Cohen[3], Tamir Ben-Hur[4], Charles L. Greenblatt[2], Hervé Bercovier[2]*

**1** Department of Urology, Hadassah- Hebrew University Medical Center, Jerusalem, Israel, **2** Department of Microbiology and Molecular Genetics, Hebrew University Jerusalem, Israel, **3** Department of Immunology, Weizmann Institute, Rehovot, Israel, **4** Department of Neurology Hadassah-Hebrew University Medical Center, Jerusalem, Israel

* hb@cc.huji.ac.il (HB); ogofrit@gmail.com (ONG)

**Data Availability Statement:** All relevant data are within the manuscript and Supporting Information files.

## Abstract

### Background

Alzheimer's disease (AD) affects one in ten people older than 65 years. Thus far, there is no cure or even disease-modifying treatment for this disease. The immune system is a major player in the pathogenesis of AD. Bacillus Calmette-Guérin (BCG), developed as a vaccine against tuberculosis, modulates the immune system and reduces recurrence of non-muscle invasive bladder cancer. Theoretical considerations suggested that treatment with BCG may decrease the risk of AD. We tested this hypothesis on a natural population of bladder cancer patients.

### Methods and findings

After removing all bladder cancer patients presenting with AD or developing AD within one-year following diagnosis of bladder cancer, we collected data on a total of 1371 patients (1134 males and 237 females) who were followed for at least one year after the diagnosis of bladder cancer. The mean age at diagnosis of bladder cancer was 68.1 years (SD 13.0). Adjuvant post-operative intra-vesical treatment with BCG was given to 878 (64%) of these patients. The median period post-operative follow-up was 8 years. During follow-up, 65 patients developed AD at a mean age of 84 years (SD 5.9), including 21 patients (2.4%) who had been treated with BCG and 44 patients (8.9%) who had not received BCG. Patients who had been treated with BCG manifested more than 4-fold less risk for AD than those not treated with BCG. The Cox proportional hazards regression model and the Kaplan-Meier analysis of AD free survival both indicated high significance: patients not treated with BCG had a significantly higher risk of developing AD compared to BCG treated patients (HR 4.778, 95%CI: 2.837–8.046, p = 4.08x$10^{-9}$ and Log Rank Chi-square 42.438, df = 1, p = 7.30x$10^{-11}$, respectively). Exposure to BCG did not modify the prevalence of Parkinson's disease, 1.9% in BCG treated patients and 1.6% in untreated (Fisher's Exact Test, p = 1).

**Funding:** Alzheimer's Germ Quest, Inc. is a privately held company, founded in 2017, and controlled by Dr. Leslie Norins and his wife, Rainey. This company provides small grants ($10,000 in our case) to investigate the idea of the possibility that Alzheimer's Disease was actually an infection of an unusual type without any request to the granted researchers. The funder had no role in study design, data collection and analysis, decision to publish, or preparation of the manuscript.

**Competing interests:** Alzheimer's Germ Quest, Inc. provided a small grant to support this study. There are no patents, products in development or marketed products associated with this research to declare. This does not alter our adherence to PLOS ONE policies on sharing data and materials.

**Abbreviations:** AD, Alzheimer's disease; PD, Parkinson's disease; Aβ, amyloid β; BCG, Bacillus Calmette-Guérin; Cox PH, Cox proportional hazards regression model; df, degrees of freedom.

## Conclusions

Bladder cancer patients treated with BCG were significantly less likely to develop AD at any age than patients who were not so treated. This finding of a retrospective study suggests that BCG treatment might also reduce the incidence of AD in the general population. Confirmation of such effects of BCG in other retrospective studies would support prospective studies of BCG in AD.

## Introduction

Alzheimer's disease (AD) affects over 5 million Americans. Its prevalence is expected to triple by 2030 and its cost to rise accordingly [1, 2]. While cardiovascular mortality was cut by more than a half in the past 30 years and cures were found for many types of cancer, no cure or even disease-modifying treatment has been found for AD. Any treatment that decreases the incidence of AD or delays its onset will have a substantial impact on the individual and on society.

AD is marked by three major pathological features: accumulation of amyloid β (Aβ) plaques, neurofibrillary tangles (hyperphosphorylated tau protein) and sustained innate neuroinflammation [3]. Since the late 1980, Aβ has been the target of most clinical trials. A series of well conducted trials with anti-Aβ agents (or with tau-aggregation inhibitors) have all failed to reduce cognitive or functional decline in patients with mild-to moderate and even "prodromal" AD [4–6]. These failures have led to a reduced effort in drug-discovery for AD by major pharmaceutical companies [7]. It seems that once dementia is present, disease progression becomes Aβ-independent and probably less susceptible to interventions [8]. In retrospect, this may not be surprising since Aβ deposition takes place years before symptoms appear.

"Setbacks in Alzheimer research demand new strategies, not surrender" is the tittle of a recent Editorial in PLOS Medicine [7]. Can modulation of the immune system and prevention, instead of treatments targeting (Aβ) or tau, be one of these new strategies?

It is now believed that inflammation is not just a response of the immune system to neuronal loss but a major player in AD pathogenesis. Neuro-inflammation contributes to neurodegeneration and to AD pathogenesis [8, 9]. The currently recognized role of the immune system in AD is as follows: the presence of Aβ activates microglia cells; the cells migrate and phagocytize the Aβ. Early in AD development, this process clears the Aβ but later the microglia are no longer able to process the Aβ. This leads them to a condition of sustained activation termed "reactive microgliosis" producing pro-inflammatory cytokines and neurotoxins. The ability of the microglia to process Aβ decreases but their state of activation persists resulting in neurotoxicity. This creates a vicious cycle of response to AD neuropathology and neurotoxicity. Toxic neuroinflammation is amplified by the recruitment of macrophages at a later stage of the disease. The immune system involvement in AD involves many classes of cells and cytokines, some promoting neurotoxicity while others are neuroprotective. It has been suggested that systemic regulatory immune cells such as CD4+ CD25+Foxp3+ (Tregs) are involved in protecting the brain from AD pathology, and their failure may contribute to disease progression [10]. Tregs regulate the immune responses critical for maintenance of self-tolerance. Their population can be expanded by administration of a relatively low dose of IL-2 [11]. Other key cytokines participating in AD include the pro-inflammatory TNFα, IL-1β and the double action cytokine IL-6, which at low levels reduces microglia activity and at high levels is pro-inflammatory [3,10]. Another prominent anti-inflammatory cytokine is IL-10. It is

released by microglia and astrocytes and inhibits the production of pro-inflammatory cyto-kines. However, clinical trials with IL-10 therapy did not support a protective role [3].

Intra-vesical Bacillus Calmette-Guérin (BCG) has been used since 1972 for preventing recurrence of non-muscle invasive bladder cancer (NMIBC). Numerous clinical trials have proved its efficacy [12]. The exact mechanism of BCG anticancer activity has not been deci-phered, but it is well recognized that BCG manifests immune effects; it binds fibronectin in the bladder wall and stimulates Th1 cells to secrete multiple cytokines including: IL-1, 2, 5, 6, 8, 10, 12 and 18, as well as IFN$\gamma$, TNF$\alpha$ and GM-CSF. It is presumed that these cytokines induce cell-mediated cytotoxic mechanisms that eliminate cancer cells.

It was recently hypothesized that administration of BCG might decrease the prevalence of AD in elderly persons through modulation of the immune system [13] possibly by expanding the Treg population [10]. This hypothesis was based on the inverse relation between BCG vac-cination against tuberculosis and the prevalence of AD/ dementia; moreover, BCG immuniza-tion has shown beneficial effects on the course of other CNS immune-related disorders like multiple sclerosis [14] and on experimental autoimmune encephalomyelitis [15]. In addition, BCG vaccination improves both brain pathology and cognitive performance in the APP/PS1 transgenic mouse model of AD [16].

In the present report, we studied the effects of BCG administration on the risk to develop AD in a population of bladder cancer patients. According to oncological guidelines, some bladder cancer patients are treated with BCG and others are not exposed to BCG. Thus, we could compare the effects of BCG on AD in these two groups. Since most cases of bladder can-cer are not acutely lethal, long-term follow-up is available in many patients.

## Methods

### Patients

The study is based on data retrieved from the computerized archives of the Hadassah-Univer-sity Hospitals. This secondary and tertiary medical center is divided into two campuses located at the opposing edges of Jerusalem serves about 1.5 million people, from both the Jewish and the Arab populations.

The archives of the Hadassah-University medical hospitals provide a unique source of long-term follow-up on many types of pathologies. The records of patients with bladder cancer were crosschecked with the hospital archive database of patients diagnosed with AD (ICD-9-cm codes 294.20 or 331.0) and Parkinson's disease (PD) (ICD-9-cm code 332.0).

The Institutional Review Board Committee (IRB number 0037-17HMO) approved the study. All data were fully anonymized before analysis and the ethics committee waived the requirement for informed consent.

### Patient management

Patients diagnosed with bladder cancer underwent transurethral resection of the tumor and an immediate instillation of a chemotherapeutic agent. Intra-vesical immunotherapy with Onco-TICE BCG 12.5mg per vial containing 2–8 x $10^8$ CFU Tice BCG was liberally used, especially in the years 1990 and 2010. During that time period BCG was offered to patients with any stage T1, any high-grade tumor, CIS, low-grade tumors larger than 3cm, more than 3 low-grade tumors and low-grade tumors that recurred within 2 years. A six-week induction course was given to these patients and maintenance doses up to three years to patients with high-grade disease [12]. For study purposes, any dose of BCG qualified the patient in the "BCG group". Intravesical chemotherapy instillation (commonly used nowadays for low-grade tumors) was rarely prescribed during the study. Follow up was maintained for five years with

periodic cystoscopies. Afterwards, annual check-up and ultrasonography of the urinary system was performed indefinitely [17].

## Statistical analysis

We used the Cox Proportional Hazards regression model (Cox PH) to estimate cause-specific hazard ratios (HRs) with 95% confidence intervals (CIs) for risk of AD comparing bladder cancer patients who did not receive BCG as a treatment to BCG treated bladder cancer patients. The effects of BCG, tumor grade, stage, and gender on the risk of AD were evaluated with the Cox PH with SPSS version 25.0 (IBM SPSS Statistics for Windows, Version 25.0. Armonk, NY: IBM Corp.). Kaplan-Meier survival analysis with the log-rank test (Mantel-Cox) were used to compare the rates of AD-free survivals in BCG treated and untreated patients. Continuous variables were compared using two-sample t-test. Fisher's Exact Test was used to compare categorical variables. All statistical tests were two tailed and a p value smaller than 0.05 was considered significant.

## Results

From 1966 to 2018, 1522 patients were treated for bladder cancer in Hadassah-University medical hospitals. For the purpose of analysis, pathological stage was dichotomized to non-muscle invasive ($<$T2) and muscle invasive ($\geq$T2) and pathological grade to low and high grades. A total of 151 patients were excluded from the analysis either because they suffered from AD while being diagnosed with bladder cancer, they died in less than one year following the diagnosis of bladder cancer or they developed AD earlier than one year after diagnosis of bladder cancer. This left 1371 patients (1134 males and 237 females) for analysis (S1 File Excel, supporting information). Clinical characteristics of the patients are presented in Table 1 and S1, S2 and S3 Tables (supporting information). Median postoperative follow-up was 8 years (IQR 3–14 years).

The mean age at diagnosis of bladder cancer was 68.1 years (SD 13.0). Patients given BCG were marginally younger than patients not given BCG (67.5±12.6 vs. 69.0±13.7, p = 0.0449). The mean age by the end of follow-up of patients given BCG was significantly older compared to patients not given (78.7±10.8 vs. 75.9±13.4, p<0.0001) reflecting the fact that BCG was not given to patients with stage $\geq$ T2. During follow-up, AD was diagnosed in 65 patients (4.7%) at a mean age of 84 years (SD 5.9), including 21 patients (2.4%) given BCG and 44 patients (8.9%) not given BCG. Among the 730 males who were given BCG, 18 were diagnosed with AD (2.47%) whereas 37 cases of AD (9.16%) were diagnosed among the 404 patient who did not receive BCG. The female population was much less represented with 7 cases of AD

**Table 1. Characteristics of patients in the observed population.**

|  | Given BCG | Not Given BCG | Total |
|---|---|---|---|
| Number of patients | 878 (64%) | 493 (36%) | 1371 |
| Mean age at diagnosis (SD) | 67.5 (12.6) | 69.0 (13.7) | 68.1 (13.0)[a] |
| Male/Female | 730/148 (83.1%/16.9%) | 404/89 (81.9%/18.1%) | 1134/237 (82.7%/17.3%) |
| Stage <T2 / ≥T2 | 867/11 (98.7%/1.3%) | 312/181 (63.3%/36.7%) | 1179/192 (86.0%/14.0%) |
| Grade Low/High | 452/426 (51.5%/48.5%) | 281/212 (57%/43%) | 733/638 (53.5%/46.5%) |
| Mean age at the end of Follow-up (SD) | 78.7 (10.8) | 75.9 (13.4) | 77.7 (11.8)[b] |
| Parkinson Yes/No | 13/865 (1.5%/98.5%) | 6/487 (1.2%/98.8%) | 19/1352 (1.4%/98.6%) |

[a]p = 0.0449
[b]p<0.0001

**Table 2. Characteristics of the 1068 patients analyzed by Cox PH and K-M.**

| | Given BCG (%within row) (%within column) | Not Given BCG (%within row) (%within column) | Total (%within row) (%within column) | p value |
|---|---|---|---|---|
| Number of patients | 702 (65.7%) | 366 (34.3%) | 1068 | |
| Mean age (SD) at AD | 82.57 (7.23) | 81.53 (6.87) | 82.2 (7.1) | 0.0231 |
| Male/Female | 587/115 (66.3%/63.2%) (83.6%/16.4%) | 299/67 (33.7%/36.8%) (81.7%/18.3%) | 886/182(65.7%/34.3%) (83.0%/17.0%) | 0.441 |
| Stage <T2 / ≥T2 | 696/8 (74.9%/5.8%) (98.9%/1.1%) | 233/131 (25.1%/(94.2%) (63.7%/35.8%) | 929/139(65.9%/34.1%) (87.0%/ 13.0%) | 1.52 x 10⁻⁵⁸ |
| Grade Low/High | 357/345 (62.9%/69.0%) (50.9%/49.1%) | 211/155 (37.1%/31.0%) (57.7%/42.3%) | 568/500(65.7%/34.3%) (53.2%/46.8%) | 0.0387 |
| Parkinson Yes/No | 13/689 (68.4%/65.7%) (1.9%/98.1%) | 6/360 (31.6%/34.3%) (1.6%/98.4%) | 19/1049(65.7%/34.3%) (1.8%/98.2%) | 1 |

(7.87%) among 89 patients non-BCG treated and 3 cases of AD (2.03%) among 148 BCG treated patients (S1, S2 and S3 Tables).

The Cox PH analysis was run for 1371 patients out of which 303 patients (22.1%) were censored cases before the earliest event of AD (occurring at age 71 years), leaving 1068 patients (93.9% censored cases) and 65 AD cases (6.1%). In the resulting 1068 patients' population, the mean age at detection of AD for BCG treated patients was 82.57 years (SD = 7.23) slightly but significantly older than 81.53 years (SD = 6.87), the mean age of the patients without BCG (p = 0.0231) (Table 2).

Cox PH analysis (93.9% censored cases) showed that BCG treatment reduced dramatically the risk of developing AD (Fig 1). This effect was highly significant (HR 4.778, 95%CI: 2.837–8.046, p = 4.08x10⁻⁹). The covariates grade and gender, were found to be not significant when run with BCG (p = 0.216 and p = 0.395 respectively) and the stage covariate was discarded as by definition patients with stage <T2 were potential candidates for BCG therapy and patients with stage ≥T2 were not. Exposure to BCG did not change the risk of Parkinson's disease (1.9% in BCG treated patients and 1.6% in untreated, p = 1 by Fisher's Exact Test, Table 2).

The Kaplan–Meier analysis performed on the population defined in the Cox PH regression (1068 patients; Table 2) showed that the AD-free survival curve of the BCG recipients was significantly different from the survival curve of non BCG treated patients (Log Rank: Chi-square

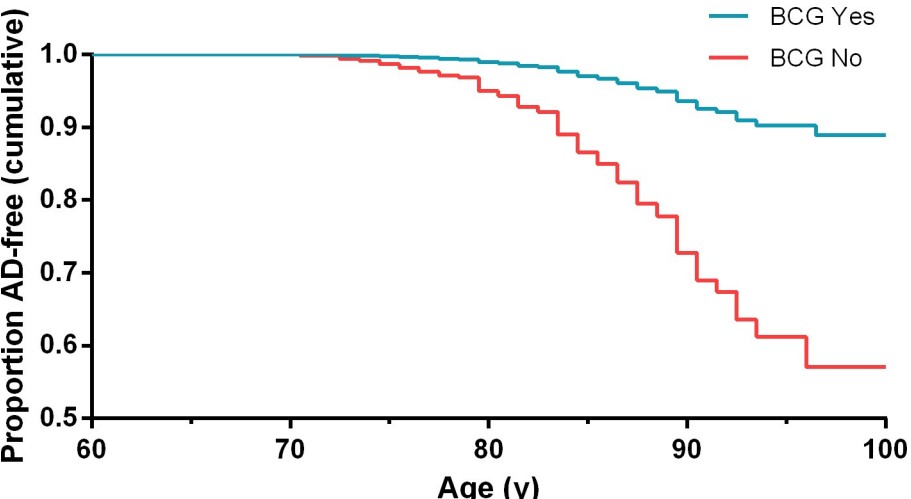

**Fig 1. Cox PH survival curves of AD-free bladder cancer patients treated or not with BCG according to age[a].** [a]HR 4.778, 95%CI: 2.837–8.046, p = 4.08x10⁻⁹.

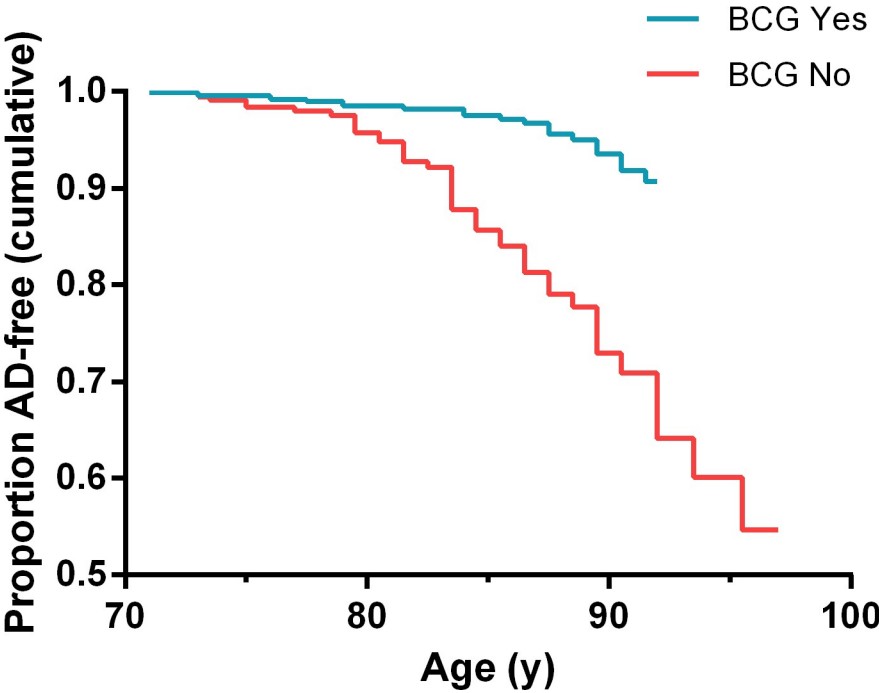

**Fig 2. Kaplan-Meier survival curves of AD-free bladder cancer patients treated or not with BCG according to age[a].** [a]Log Rank Chi-square 42.438, df = 1, p = 7.30E-11 between BCG treated and BCG not treated patients (93.9% of the 1068 patients were censored cases).

42.438,df = 1, p = $7.30 \times 10^{-11}$) with overall 93.9% censored cases (Fig 2). When stratifying the population by gender, the difference remained highly significant (males: Log Rank Chi-Square of 35.162, df = 1 and p = $3.03 \times 10^{-9}$; females: Log Rank Chi-Square of 6.735, df = 1 and p = 0.00945) (S1 and S2 Figs). The higher p value for females is probably due to the small size of the female population. There were no significant differences in the AD-free survival between male and female patients not given BCG (Log Rank: Chi-square 0.390, df = 1, p = 0.533). This was also true between BCG treated females and males (Log Rank: Chi-square 0.433, df = 1, p = 0.511) (S3 and S4 Figs). The Cohen effect sizes [$w = \sqrt{(\chi 2/N)}$)] were large and similar, 0.82 and 0.80 for females and males respectively. This result lends support to the finding from the Cox PH model that gender is not a significant co-variate (p = 0,395) and mitigates against any marked introduction of bias caused by combining males and females data in the general analysis [18].

## Discussion

Bladder cancer patients typically present in their 6th to 8th decades of life making them an appropriate population for investigations in AD. In this study, we followed a cohort of these patients for a median period of 8 years after diagnosis of bladder cancer. About 60% of them were given adjuvant treatment with intravesical instillations of BCG and the rest were not. During follow-up, 4.7% of the patients developed AD, a rate similar to the 4.66% reported in a recent European meta-analysis [19]. Cox PH and Kaplan-Meier analyses documented respectively that patients not treated with BCG had a significantly higher risk of developing AD and a significantly different AD-free survival (HR 4.778, 95%CI: 2.837–8.046, p = $4.08 \times 10^{-9}$ and Log Rank Chi-square 42.438, df = 1, p = $7.30 \times 10^{-11}$) compared to BCG treated patients (Figs 1

and [2]). This happened despite the significantly older age of patients given BCG (Table 1). Exposure to BCG did not change the risk of developing Parkinson's disease.

The incidence of AD (after deleting all the patients censored before the first case of AD; Table 2) rose to 6%, which is in the range reported in the general population at this age [19].

Male and Female populations differ in both the incidence of bladder cancer and AD. AD affects women more commonly while bladder cancer is more common in men [20]. Stratifying the population for gender, we found a highly significant effect of BCG in both men (Log Rank Chi-Square of 35.162, df = 1, p = $3.03 \times 10^{-9}$) and women AD-free survival (Log Rank Chi-Square of 6.735, df = 1, p = 0.00945) (S1 and S2 Figs). AD incidence was not higher in females compared to males (10/182 = 5.5% vs 55/886 = 6.2%). This may indicate that this population is unique. However, the small number of female patients and likewise the small number of AD cases among females preclude a definite conclusion.

BCG, the tuberculosis vaccine, has been used by urologists for many years as an anti-cancer medication, but it also appears to show activity against other diseases. It is well recognized nowadays that immunization with living organisms has what is termed "off-target" effects, which in different populations can be dramatic. Aaby et al., in a randomized trial conducted in Guinea-Bissau, showed that immunization with BCG reduced neonatal mortality in low birth-weight neonates by 50% [21]. This effect is not fully understood, but induction of innate immune memory and heterologous lymphocyte activation have been mentioned as possible explanations [22].

How can an anti-cancer agent instilled into the bladder exert an impact outside the bladder? Does BCG given intravesically provoke a systemic immune response? The answer is yes; a ten-fold rise in the serum levels of IL-2 and a fivefold increase in IFNγ were documented at 5–6 weeks after intravesical BCG instillation. This was accompanied by a threefold increase in BCG induced killer-cell activity manifested by peripheral blood mononuclear cells [23].

Is there any other evidence connecting BCG immunization to AD prevention? Yes, there are epidemiological observations as well as experiments in an animal model. A strong inverse association was found between AD prevalence and BCG vaccination in countries where it is routinely used [13]. BCG administration was stopped in the U.S. in the early 60's. Might this have contributed to the rise in AD we currently notice? [13]. APP/PS1 mice provide a transgenic model of AD. These mice develop the deposition of amyloid plaques in the brain at six weeks of age. BCG immunization inhibits both brain pathology and cognitive dysfunction in this model [16]. BCG treatment even reversed cognitive decline. BCG-treated APP/PS1 mice showed recruitment of inflammation-resolving monocytes across the choroid plexus and perivascular spaces to cerebral sites of plaque pathology, increased circulating IFNγ levels, upregulation of cerebral anti-inflammatory cytokine levels and elevated expression of neutrophic factors in the brain. In closer proximity to bladder cancer, a recent large Surveillance, Epidemiology, and End Results (SEER) based study reviewed the records of 23, 932-bladder cancer patients [24]. They found that BCG treated patients had better disease specific survival (HR, 0.9; CI, 0.8 to 1.01), but the effect of BCG on overall survival was even greater (HR, 0.87; CI, 0.83 to 0.92). The authors acknowledge that this could be due to an "unmeasured factor" and could represent an "off target" effect of BCG.

How might BCG given intravesically decrease the occurrence of AD? The exact mechanism of the BCG anti AD effect is obviously unknown. It is however, well appreciated that intra-vesical BCG increases systemic IL-2 levels [23], which expands the populations of the neuroprotective Treg cells [11] In addition, BCG increases anti-inflammatory cytokines in the brain and therefore reduces neuro-inflammation which is one of the three major pathological features of AD [9, 16]. In the APP/PS1 mouse model, amplification of these cells by peripheral IL-2 decreased plaque formation and restored cognitive function [16]. Changes in the blood-

cerebrospinal fluid barrier found in patients with mild cognitive impairment may facilitate the entry of systemic IL-2 into the nervous system [25]. Neonatal BCG vaccination has been shown to induce on the one hand anti-inflammatory meningeal macrophage M2 polarization and neurotrophic factor expression that were T lymphocytes and Il-10 dependent and on the other hand neuroprotective Tregs in models of neurodegenerative diseases [10, 26, 27].

Does this effect result from the direct activity of BCG on the immune system or indirectly by inducing changes in the microbiota? To the best of our knowledge, there is no definitive data on the effect of BCG on the microbiota in animals or in man. Some data on the bladder ecosystem were recently reported pointing to the importance of Proteobacteria in bladder cancer recurrence [28]. It has been shown in a Specific Pathogen Free (SPF) APP/PS1 transgenic mouse model that there is a shift in the gut microbiota with an increase of Proteobacteria compared to non-transgenic wild-type mice [29]. In addition, Germ Free APP/PS1 mice did not seem to develop the same level of Aβ amyloid as their SPF counterpart. When repopulated with the microbiota of the conventionally raised APP/PS1 transgenic mice, their cerebral level of Aβ amyloid increased substantially [30].

### Limitations of the study

The study presented here has several potential confounding caveats:

1. Selection bias. More fit patients may have been given BCG more often than were the less fit patients. One cannot rule out the possibility that the treating urologist ruled out BCG treatment in the frailer (and possibly more prone to AD) patients.

2. The large differences in the AD risk of bladder cancer patients reported here and in the USA may be due to the fact that 60% of the bladder cancer patients at Hadassah Hospital received BCG whereas in the USA probably less than 20% of all bladder cancer patients received BCG [31], masking its effect in large retrospective studies.

3. Bias due to high percentage of censored data. Only 4.7% of the patients developed AD in the observed population meaning that 95.3% of the cases were censored in the Cox PH analysis. This number declined to 93.9% in the Kaplan-Meier analysis after removing the 303 cases that were censored at an age younger than the age of the first AD event (71 years). A high percentage of censored events may increase the risk of bias [32]. On the other hand, the Kaplan-Meier survival curves never crossed each other mitigating against bias from this source (Figs 1 and 2).

4. Bias due to a lower risk of AD in cancer survivors. Several reports including a large cohort study and two prospective studies showed that the risk of AD is reduced by 33–35% in cancer patients [33, 34]. In one study, late-stage cancer was associated with an even lower risk of AD (HR = 0.50, 95%CI: 0.27, 0.94) [35]. This may result from metabolic differences between AD and cancer patients: Cancer = upregulation of glycolysis; AD = upregulation of oxidative phosphorylation [36]. The beneficial effect of BCG in our study was much stronger: HR = 4.778 for AD risk in non-BCG treated patients compared to treated patients. Since all patients in the current study were cancer survivors, we can hypothesize that BCG had a significant additive role in lowering the AD risk. Still one cannot rule out the possibility that BCG survivors have a yet unexplained reduced risk for AD that is not directly due to the effect of BCG.

5. Bias due to the retrospective design of the study and reliance on the ICD coding system. The diagnosis of AD in the current study was based on hospital ICD coding and not on more accurate measures such as mental tests or imaging studies. This shortcoming should

act similarly on both recipients and non-recipients of BCG and should not modify the results largely.

6. Bias due to the different sensitivity for diagnosis of AD in different hospital departments. Diagnosing AD is often difficult especially in emergency admissions or when patient's history is not available. Some AD cases could have been missed in this way. This bias should also act similarly on recipients and non-recipients of BCG.

7. Men and women presented similar rates of incidence of AD. This differs from the standard reported higher rate of AD in women. It could be due to the lesser representation of women in the study but could also suggest some other unique characteristics of the studied population.

8. A dose-dependent relationship between the number of BCG instillations and the risk of AD was not established. This was due to the small number of patients with AD in the BCG group. The absence of dose-dependent relations decreases the strength of our conclusions.

9. A Mantoux test or other tests of lymphocyte responsiveness to BCG were not done. These tests are not part of the routine management of bladder cancer patients today. Therefore, we do not know which patient responded immunologically to BCG.

In summary, we showed that BCG given intravesically to patients with bladder cancer lowered their risk of developing AD by more than fourfold. This is only a single observation found retrospectively in a unique population. There is however, a plausible biological explanation for this phenomenon and supporting findings are found in epidemiological studies and in an animal model of AD. We hope that the results presented here will stimulate studies in other populations and further work on the mechanism of BCG protection will be done. A prospective BCG vaccination study of elderly subjects can then be envisioned with different doses of BCG and a continuous follow-up of cognitive capacities. Obviously, a preventive strategy would not be via a bladder administered vaccination. The intra dermic route or an oral route with an appropriate formulation should be considered [37].

## Supporting information

**S1 Table. AD and Age distribution of patients (Male and Female) not treated or treated with BCG.**
(DOCX)

**S2 Table. AD and Age distribution of patients (Male only) not treated or treated with BCG.**
(DOCX)

**S3 Table. AD and Age distribution of patients (Female only) not treated or treated with BCG.**
(DOCX)

**S1 Fig. Kaplan–Meier survival curves of the AD-free female patients according to treatment (BG vs. No BCG) and to age[a].** [a]Log Rank: Chi-Square 6.735, df = 1, p = 0.00945.
(DOCX)

**S2 Fig. Kaplan–Meier survival curves of the AD-free male patients according to treatment (BCG vs. No BCG) and to age[a].**
(DOCX)

**S3 Fig. Kaplan–Meier survival curves of the AD-free male and female patients not treated with BCG[a].** [a]Log Rank: Chi-Square 35.162, df = 1, p = $3.03 \times 10^{-9}$.
(DOCX)

**S4 Fig. Kaplan–Meier survival curves of the AD-free male and female patients treated with BCG[a].** [a]Log Rank: Chi-square 0.433, df = 1, p = 0.511.
(DOCX)

**S1 File. Excel.**
(XLSX)

## Acknowledgments

We wish to thank a member of the Department of Developmental Biology and Cancer Research, Faculty of Medicine, Hebrew University of Jerusalem, who advised us on sampling size, performed the Kaplan-Meier and Cox Proportional Hazards analyses and reviewed the article. We thank Naseem Maalouf for the preparation of the graphs done on GraphPad Prism software version 6.0.

## Author Contributions

**Conceptualization:** Ofer N. Gofrit, Benjamin Y. Klein, Irun R. Cohen, Tamir Ben-Hur, Charles L. Greenblatt, Hervé Bercovier.

**Data curation:** Ofer N. Gofrit.

**Formal analysis:** Ofer N. Gofrit, Charles L. Greenblatt, Hervé Bercovier.

**Funding acquisition:** Charles L. Greenblatt.

**Investigation:** Ofer N. Gofrit, Charles L. Greenblatt, Hervé Bercovier.

**Methodology:** Ofer N. Gofrit, Hervé Bercovier.

**Project administration:** Ofer N. Gofrit, Hervé Bercovier.

**Resources:** Ofer N. Gofrit.

**Software:** Ofer N. Gofrit, Hervé Bercovier.

**Supervision:** Ofer N. Gofrit, Hervé Bercovier.

**Validation:** Ofer N. Gofrit, Hervé Bercovier.

**Visualization:** Ofer N. Gofrit, Hervé Bercovier.

**Writing – original draft:** Ofer N. Gofrit, Hervé Bercovier.

**Writing – review & editing:** Ofer N. Gofrit, Benjamin Y. Klein, Irun R. Cohen, Tamir Ben-Hur, Charles L. Greenblatt, Hervé Bercovier.

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
