## [Decision Letter · Decision Letter 0]

19 Sep 2019

PONE-D-19-23671

Bacillus Calmette-Guérin (BCG) therapy lowers the incidence of Alzheimer’s disease in bladder cancer patients

PLOS ONE

Dear Prof. Gofrit,

Thank you for submitting your manuscript to PLOS ONE. After careful consideration by two Reviewers and an Academic Editor, please make the suggested corrections posed by the Reviewers so I can render a decision on this manuscript.

**Comments to the Author**

1. Is the manuscript technically sound, and do the data support the conclusions?

Reviewer #1: Yes

Reviewer #2: Yes

2. Has the statistical analysis been performed appropriately and rigorously? 

Reviewer #1: I Don't Know

Reviewer #2: Yes

3. Have the authors made all data underlying the findings in their manuscript fully available?

Reviewer #1: Yes

Reviewer #2: Yes

4. Is the manuscript presented in an intelligible fashion and written in standard English?

Reviewer #1: Yes

Reviewer #2: Yes

5. Review Comments to the Author

Reviewer #1: The authors reported that Bacillus Calmette-Guérin (BCG) could decrease the risk of Alzheimer’s disease, through modulating immune system. The observed data are from a total of 1371 bladder cancer patients. It is very interesting, and will provide valuable clue for immunotherapy in Alzheimer’s disease.

1. Due to the different sensitivity for bladder cancer patients, BCG therapy will fail in 37% to 45% of patients treated over a 2-year span (PMID: 28801026). Is this affect the conclusion in the present study ? Could you give results of cytokine levels in the BCG treated patients? Such as IFN-gamma, IL-2, TNF-a, chemotactic factors and so on.

2. The dose and injection times of BCG should be presented and analyzed in this study.

3. Could you add some potential mechanisms for BCG-related reducation of AD risk in bladder cancer patients in the disscussion section ?

Reviewer #2: This study was conducted in 1371 non-muscle invasive bladder cancer patients with or without BCG treatment followed by long time follow up. The data reveal a significant benefit of BCG treatment in bladder cancer patients in prevention of AD development. The findings and conclusion are clear and straight forward. Moreover, the discussion and statistical analysis are complete and comprehensive.

1. The findings support current understanding on the beneficial effects of BCG in prevention of AD development.

2. The figure legends of Fig. 1 and 2 are not shown.

3. A few grammar errors: e.g. p. 11 line 105 “effects and on the course..”

6. PLOS authors have the option to publish the peer review history of their article (what does this mean?). If published, this will include your full peer review and any attached files.

**Do you want your identity to be public for this peer review?** For information about this choice, including consent withdrawal, please see our Privacy Policy.

Reviewer #1: Yes: Zhibin Yao and Fangfang Qi

Reviewer #2: No

We would appreciate receiving your revised manuscript by December, 2020. To enhance the reproducibility of your results, we recommend that if applicable you deposit your laboratory protocols in protocols.io, where a protocol can be assigned its own identifier (DOI) such that it can be cited independently in the future. For instructions see: http://journals.plos.org/plosone/s/submission-guidelines#loc-laboratory-protocols

We look forward to receiving your revised manuscript.

Kind regards,

Stephen D. Ginsberg, Ph.D.

Section Editor

PLOS ONE

2. Our internal editors have looked over your manuscript and determined that it may be within the scope of our Early Diagnosis and Treatment of Alzheimer's Disease Call for Papers. This collection of papers is headed by a team of Guest Editors for PLOS ONE: Michael Weiner, Roberta Brinton, Jussi Tohka and Yona Levites. With this Collection we hope to bring together researchers working on a wide range of disciplines, from molecular and preclinical work, through to patient-centered studies, including clinical trials.   Additional information can be found on our announcement page: https://collections.plos.org/s/alzheimersdisease. If you would like your manuscript to be considered for this collection, please let us know in your cover letter and we will ensure that your paper is treated as if you were responding to this call. Agreeing to be part of the call-for-papers will not affect the date your manuscript is published. If you would prefer to remove your manuscript from collection consideration, please specify this in the cover letter.

3. In the ethics statement in the manuscript and in the online submission form, please provide additional information about the patient records used in your retrospective study. Specifically, please ensure that you have discussed whether all data were fully anonymized before you accessed them and/or whether the IRB or ethics committee waived the requirement for informed consent. If patients provided informed written consent to have data from their medical records used in research, please include this information.

4. Thank you for stating the following in the Competing Interests/Financial Disclosure * (delete as necessary) section:

"CLG got a grant from the Alzheimer's Germ Quest, Inc. https://alzgerm.org. The funders had no role in study design, data collection and analysis, decision to publish, or preparation of the manuscript."  

We note that you received funding from a commercial source: [Name of Company]

---

## [Author Response · Author response to Decision Letter 0]

8 Oct 2019

Stephen D. Ginsberg, Ph.D.

Section Editor

PLOS ONE

Dear Editor 

Thank you for your letter and the comments regarding PONE-D-19-23671, ”Bacillus Calmette-Guérin (BCG) therapy lowers the incidence of Alzheimer’s disease in bladder cancer patients” and for the opportunity to submit a revised version. Here are our responses to the comments:

Comment 1. Is the manuscript technically sound, and do the data support the conclusions?

Reviewer #1: Yes

Reviewer #2: Yes

Response: We wish to thank the reviewers.

Comment 2. Has the statistical analysis been performed appropriately and rigorously? 

Reviewer #1: I Don't Know

Reviewer #2: Yes

Response: We wish to thank the reviewers.

Comment 3. Have the authors made all data underlying the findings in their manuscript fully available?

Reviewer #1: Yes

Reviewer #2: Yes

Response: We wish to thank the reviewers.

Comment 4. Is the manuscript presented in an intelligible fashion and written in standard English?

Reviewer #1: Yes

Reviewer #2: Yes

Response: We wish to thank the reviewers.

Comment 5. Review Comments to the Author

Reviewer #1: The authors reported that Bacillus Calmette-Guérin (BCG) could decrease the risk of Alzheimer’s disease, through modulating immune system. The observed data are from a total of 1371 bladder cancer patients. It is very interesting, and will provide valuable clue for immunotherapy in Alzheimer’s disease.

Comment 1. Due to the different sensitivity for bladder cancer patients, BCG therapy will fail in 37% to 45% of patients treated over a 2-year span (PMID: 28801026). Is this affect the conclusion in the present study ? Could you give results of cytokine levels in the BCG treated patients? Such as IFN-gamma, IL-2, TNF-a, chemotactic factors and so on.

Response: Indeed, the response to BCG is not complete and there are patients with disease recurrence or progression after BCG and nobody knows if it relates to the immune response per se to BCG. Nevertheless, as shown in the graph below, the overall survival of patient given BCG was significantly better (p<0.0001 Log-Rank test) than that of patients not given (Fig. #1 below). This was mainly due to better disease specific survival (p<0.0001 Log-Rank test, Fig. #2, below), since patients with muscle-invasive disease were not candidate for BCG. The final result is that patients given BCG were older at the end of the observational period (mean 78.8±10.8 vs. 75.9±13.4, p=0.0005) and therefore, were at a higher risk of developing AD compared to untreated patients. This important point strengthens the conclusions and was added to the results section (to Table 1) and to the discussion. As our study was retrospective (starting 20 years ago), cytokine levels were never evaluated in bladder cancer patients. In future prospective studies, the determination of levels of relevant cytokines should definitely be included.

Fig.#1

 

Fig. #2

Comment 2. The dose and injection times of BCG should be presented and analyzed in this study.

Response: OncoTICE BCG 12.5mg per vial containing 2-8 x 108 CFU Tice BCG was used throughout the study. This was added to the “patient’s management” section. treatment schedule is presented in this section.

Comment 3. Could you add some potential mechanisms for BCG-related reduction of AD risk in bladder cancer patients in the discussion section?

Response: We added two references [25 and 26] to the discussion of the potential impact of BCG on AD course. Laćan et al (ref #26) showed that BCG induces Tregs cells. In addition, BCG increases anti-inflammatory cytokine levels in the brain and therefore reduces neuro-inflammation which is one of the three major pathological features of AD [16]. Furthermore, neonatal BCG vaccination has been shown on one hand to induce anti-inflammatory meningeal macrophage M2 polarization and neurotrophic factor expression that were T lymphocytes and Il-10 dependent and on the other hand neuroprotective Tregs in models of neurodegenerative diseases [ref #10, 25, 26]. 

Reviewer #2: This study was conducted in 1371 non-muscle invasive bladder cancer patients with or without BCG treatment followed by long time follow up. The data reveal a significant benefit of BCG treatment in bladder cancer patients in prevention of AD development. The findings and conclusion are clear and straight forward. Moreover, the discussion and statistical analysis are complete and comprehensive.

Comment 1. The findings support current understanding on the beneficial effects of BCG in prevention of AD development.

Response: We wish to thank the reviewer.

Comment 2. The figure legends of Fig. 1 and 2 are not shown.

Response: The legends appear below the figures (as requested for an initial draft). We will change this in the final submission. 

Comment 3. A few grammar errors: e.g. p. 11 line 105 “effects and on the course.”

Response: A thorough revision of the manuscript by a native English speaker (Irun R. Cohen) was done again.

6. PLOS authors have the option to publish the peer review history of their article (what does this mean?). If published, this will include your full peer review and any attached files.

Do you want your identity to be public for this peer review? For information about this choice, including consent withdrawal, please see our Privacy Policy.

Reviewer #1: Yes: Zhibin Yao and Fangfang Qi

Reviewer #2: No

Response: This was done.

We would appreciate receiving your revised manuscript by December, 2020 !!!!. Response: See below what is stated about the funding body.

To enhance the reproducibility of your results, we recommend that if applicable you deposit your laboratory protocols in protocols.io, where a protocol can be assigned its own identifier (DOI) such that it can be cited independently in the future. For instructions see: http://journals.plos.org/plosone/s/submission-guidelines#loc-laboratory-protocols

• A rebuttal letter that responds to each point raised by the academic editor and reviewer(s). This letter should be uploaded as separate file and labeled 'Response to Reviewers'.

• A marked-up copy of your manuscript that highlights changes made to the original version. This file should be uploaded as separate file and labeled 'Revised Manuscript with Track Changes'.

• An unmarked version of your revised paper without tracked changes. This file should be uploaded as separate file and labeled 'Manuscript'.

We look forward to receiving your revised manuscript.

Kind regards,

Stephen D. Ginsberg, Ph.D.

Section Editor

PLOS ONE

Response: This was done

2. Our internal editors have looked over your manuscript and determined that it may be within the scope of our Early Diagnosis and Treatment of Alzheimer's Disease Call for Papers. This collection of papers is headed by a team of Guest Editors for PLOS ONE: Michael Weiner, Roberta Brinton, Jussi Tohka and Yona Levites. With this Collection we hope to bring together researchers working on a wide range of disciplines, from molecular and preclinical work, through to patient-centered studies, including clinical trials. Additional information can be found on our announcement page: https://collections.plos.org/s/alzheimersdisease. If you would like your manuscript to be considered for this collection, please let us know in your cover letter and we will ensure that your paper is treated as if you were responding to this call. Agreeing to be part of the call-for-papers will not affect the date your manuscript is published. If you would prefer to remove your manuscript from collection consideration, please specify this in the cover letter.

Response: We agree for including the manuscript in the collection.

3. In the ethics statement in the manuscript and in the online submission form, please provide additional information about the patient records used in your retrospective study. Specifically, please ensure that you have discussed whether all data were fully anonymized before you accessed them and/or whether the IRB or ethics committee waived the requirement for informed consent. If patients provided informed written consent to have data from their medical records used in research, please include this information.

Response: All data were fully anonymized before analysis and the ethics committee waived the requirement for informed consent. This was added to the “Methods”.

4. Thank you for stating the following in the Financial Disclosure section:

Response: Charles L. Greenblatt got a grant from the Alzheimer's Germ Quest, Inc. https://alzgerm.org. The funders had no role in study design, data collection and analysis, decision to publish, or preparation of the manuscript." 

We note that you received funding from a commercial source: [Name of Company]

Response: Alzheimer’s Germ Quest, Inc. is a privately held company, founded in 2017, and controlled by Dr. Leslie Norins and his wife, Rainey. This company provides small grants (10,000 $ in our case) to investigate the idea of the possibility that AD was actually an infection of an unusual type without any request to the granted researchers. It is incorporated in Florida, as a public benefit corporation. “Public benefit” indicates that one of its charter purposes is to help the citizenry, in this case accelerating and intensifying the search for infectious agents as root causes of AD. This firm is completely independent and not affiliated with or endorsed by any government agency, nonprofit group, pharmaceutical company, or other entity. Donations from the public are neither solicited nor accepted. The firm’s capital and operating funds come from its founders. Therefore, there is no Competing Interests and this does not alter our adherence to PLOS ONE policies on sharing data and materials.

Response: This was done.

Response: As stated, there are no Competing Interests.

Please know it is PLOS ONE policy for corresponding authors to declare, on behalf of all authors, all potential competing interests for the purposes of transparency. PLOS defines a competing interest as anything that interferes with, or could reasonably be perceived as interfering with, the full and objective presentation, peer review, editorial decision-making, or publication of research or non-research articles submitted to one of the journals. Competing interests can be financial or non-financial, professional, or personal. Competing interests can arise in relationship to an organization or another person. Please follow this link to our website for more details on competing interests: http://journals.plos.org/plosone/s/competing-interests.

Finally, our colleague who performed most of the statistics is an Emeritus Professor who does not want to appear in publications in general. Therefore, we added him (with his agreement) in the Acknowledgments. His last request, after seeing that the paper was under revision, was to remain anonymous and therefore we changed slightly the Acknowledgments to respect his will as follow: 

We wish to thank a member of the Department of Developmental Biology and Cancer Research, expert in mathematical biological models, Faculty of Medicine, Hebrew University of Jerusalem, who advised us on sampling size, performed the Kaplan-Meier and Cox Proportional Hazards analyses and reviewed the article. We thank Naseem Maalouf for the preparation of the graphs done on GraphPad Prism software version 6.0.

Sincerely

Ofer N. Gofrit and Herve Bercovier

---

## [Editor Report · Decision Letter 1]

15 Oct 2019

Bacillus Calmette-Guérin (BCG) therapy lowers the incidence of Alzheimer’s disease in bladder cancer patients

PONE-D-19-23671R1

Dear Dr. Gofrit,

We are pleased to inform you that your manuscript has been judged scientifically suitable for publication and will be formally accepted for publication once it complies with all outstanding technical requirements.

With kind regards,

Stephen D. Ginsberg, Ph.D.

Section Editor

PLOS ONE

---

## [Editor Report · Acceptance letter]

30 Oct 2019

PONE-D-19-23671R1 

Bacillus Calmette-Guérin (BCG) therapy lowers the incidence of Alzheimer’s disease in bladder cancer patients 

Dear Dr. Gofrit:

I am pleased to inform you that your manuscript has been deemed suitable for publication in PLOS ONE. Congratulations! Your manuscript is now with our production department. 

With kind regards,

on behalf of

Dr. Stephen D Ginsberg 

Section Editor

PLOS ONE